# Metagenomic Insights into the Regulatory Effects of Microbial Community on the Formation of Biogenic Amines and Volatile Flavor Components during the Brewing of *Hongqu* Rice Wine

**DOI:** 10.3390/foods12163075

**Published:** 2023-08-16

**Authors:** Ziyi Yang, Wenlong Li, Yujie Yuan, Zihua Liang, Yingyin Yan, Ying Chen, Li Ni, Xucong Lv

**Affiliations:** Food Nutrition and Health Research Center, School of Advanced Manufacturing, Fuzhou University, Jinjiang 362200, China; 218527278@fzu.edu.cn (Z.Y.); 208527253@fzu.edu.cn (W.L.); 210820055@fzu.edu.cn (Y.Y.); 220820089@fzu.edu.cn (Z.L.); 218527276@fzu.edu.cn (Y.Y.); 210820067@fzu.edu.cn (Y.C.); nili@fzu.edu.cn (L.N.)

**Keywords:** *Hongqu* rice wine, microbial community, metagenomics, biogenic amines, volatile flavor components

## Abstract

As one of the typical representatives of Chinese rice wine (*Huangjiu*), *Hongqu* rice wine is produced with glutinous rice as the main raw material and *Hongqu* as the fermentation starter. The complex microbial flora in the brewing process may have a great influence on the formation of the flavor quality and drinking safety of *Hongqu* rice wine. Previous studies have shown that high biogenic amine (BA) content in rice wine has potential physiological toxicity and has become a bottleneck problem restricting the development of the rice wine industry. This study aimed to evaluate the regulatory effects of the microbial community on the formation of BAs and volatile flavor components during the brewing of *Hongqu* rice wine. The results demonstrated that histamine, putrescine, cadaverine, tyramine, tryptamine, spermine, and spermidine were the main BAs in *Hongqu* rice wine. The contents of putrescine, cadaverine, histamine, tyramine, and spermidine in *Hongqu* rice wine of HBAs (with higher BAs content) were significantly higher than those in LBAs (with lower BAs content). GC-MS testing results showed that there were significant differences in the composition of volatile organic compounds (VOCs) between HBAs and LBAs. Among them, VOCs such as 2-methoxy-4-vinylphenol, ethyl caprate, phenethyl acetate, ethyl lactate, ethyl myristate, ethyl palmitate, ethyl n-octadecanoate, ethyl oleate, and ethyl linoleate were identified as the characteristic volatile components with significant differences between HBAs and LBAs. Microbiome analysis based on metagenomic sequencing revealed that unclassified_g_*Pantoea*, *Klebsiella pneumoniae*, *Panobacter disperse*, unclassified_f_*Enterobacteriaceae*, *Leuconostoc mesenteroides*, and *Saccharomyces cerevisiae* were the dominant microbial species in the HBA brewing process, while *Weissella confuse*, *Pediococcus acidilactici*, *Saccharomyces cerevisiae*, and *Aspergillus niger* were the dominant microbial species in the LBA brewing process. Furthermore, correlation heatmap analysis demonstrated that BAs were positively related to *Lactobacillus curvatus*, *Lactococcus lactis*, and *Leuconostoc mesenteroides*. Bioinformatical analysis based on the KEGG database revealed that the microbial genes encoding enzymes involved in BAs’ synthesis were more abundant in HBAs, and the abundances of microbial genes encoding enzymes related to BAs’ degradation and the metabolism of characteristic volatile components were higher in LBAs. Overall, this work provides important scientific data for enhancing the flavor quality of *Hongqu* rice wine and lays a solid foundation for the healthy development of the *Hongqu* rice wine industry.

## 1. Introduction

Chinese rice wine (*Huangjiu*), a typical representative of traditional fermented foods in China, is one of the oldest traditionally brewed wines in the world. It occupies an important position among the three major brewed wines. As a typical representative of Chinese rice wine (also known as *Huangjiu*), *Hongqu* rice wine is brewed with glutinous rice as the main raw material and *Hongqu* (rich in *Monascus* spp.) as the fermentation starter. Due to the addition of *Hongqu*, the resulting *Hongqu* rice wine has an attractive reddish color, unique flavor, and potential health benefits, because *Monascus* spp. can produce a variety of functional secondary metabolites, such as *Monascus* pigments, monacolin K, γ-aminobutyric acid (GABA), etc. [1]. At present, the production of *Hongqu* rice wine is usually carried out in an open non-sterile environment based on empirical knowledge. The succession and metabolic activities of complex microbial community shape the unique flavor and quality of *Hongqu* rice wine. In the traditional brewing process of *Hongqu* rice wine, some undesirable microorganisms may produce metabolites that have adverse effects on human health, such as biogenic amines, higher alcohols, mycotoxins, etc., which seriously hinders the healthy development and internationalization of the *Hongqu* rice wine industry.

Biogenic amines (BAs) are a class of toxic low-molecular weight, nitrogen-containing organic compounds that are widely present in various fermented foods, such as *Huangjiu*, wine, vinegar, fermented vegetables, and soy sauce [2]. Based on their chemical structures, BAs can be divided into three categories: aliphatic amines (e.g., putrescine, cadaverine, spermine, and spermidine), aromatic amines (e.g., phenylethylamine and tyramine), and heterocyclic amines (e.g., tryptamine and histamine) [3]. Moderate intake of BAs is generally considered beneficial as they can promote growth, enhance metabolism, eliminate free radicals [4], and serve as precursors for the synthesis of biologically active substances such as hormones, alkaloids, and proteins [5]. However, high concentrations of BAs may have adverse effects on the body, such as nausea, migraines, respiratory distress, palpitations, and abnormal blood pressure, and in severe cases can lead to cerebral hemorrhage and heart failure [5,6]. The presence of BAs in fermented wines is an important food safety issue that can no longer be ignored. Previous studies have shown that the total BAs content in Chinese rice wine is generally high, with an average concentration of up to 115 mg/L [7], much higher than that of other alcoholic fermented beverages such as *Baijiu* (5.35 mg/L) [8], beer (39.6 mg/L) [9], and wine (45.5 mg/L) [10]. The most-frequently observed BAs in Chinese rice wine are putrescine, tyramine, histamine, cadaverine, spermine, spermidine, phenylethylamine, and tryptamine. For *Hongqu* rice wine, our previous study showed that the content of biogenic amines can reach more than 80 mg/L [11]. Furthermore, alcohol in Chinese rice wine would inhibit the activity of amine oxidase in the human body, further increasing the toxicity of biogenic amines [12]. Therefore, the high BAs content in *Hongqu* rice wine may pose a health threat to the drinker and lead to a poor post-drinking experience, typically characterized by migraine [13], which has become a bottleneck problem restricting the healthy development and internationalization of the *Hongqu* rice wine industry. However, the microbial species involved in BAs’ metabolism and their metabolic pathways in the traditional brewing process of *Hongqu* rice wine are still unclear, which hinders the regulation of the abatement of BAs content in *Hongqu* rice wine. 

Previous studies have preliminarily revealed the potential relationship between microbial community and BAs in the brewing of Chinese rice wine. For example, a previous study based on the partial least-squares regression model showed that the formation of BAs in Shaoxing rice wine was highly correlated with *Staphylococcus*, *Leuconostoc*, *Weissella*, and *Lactococcus*, but was not significantly correlated with *Bacillus*, *Lactobacillus*, *Pseudomonas*, and *Thermoactinomyces* [14]. In another previous study, *Lactobacillus*, *Bacillus*, *Pseudomonas*, and *Enterobacter* spp. were reported as important contributors to the decarboxylase gene family and were positively associated with BAs’ formation in Chinese rice wine, but the BAs-producing bacteria were strain-specific [15]. Our previous study showed that the formation of spermidine, cadaverine, and putrescine in the *Hongqu* rice wine brewing process was positively correlated with *Lactobacillus*, *Vanrija*, *Apiotrichum*, *Millerozyma*, etc., based on Spearman correlation analysis [11]. However, the potential relationship between BAs’ formation and microorganisms was mostly predicted based on amplicon high-throughput sequencing, as well as unvalidated statistical methods. The analysis of potential biogenic amine-producing bacteria can only be performed at the genus level, so the results of these studies may be contradictory with respect to the actual situation. Therefore, the functional microbial species involved in BAs’ formation during the fermentation of *Hongqu* rice wine are still unclear. Metagenomics is considered as an ideal tool to capture the metabolic function of microbial communities in complex environments, which has been widely used to elucidate the dynamic changes of microbial community and gene function under complex environmental conditions of traditional fermented foods such as Zhenjiang aromatic vinegar [16], *Daqu* [17], fermented milk [18], and sausages [19]. Metagenomics can be used to deeply understand the microbial species source of key genes in the brewing environment from the perspective of biological correlation, so as to reveal the metabolic potential of microbial flora. Therefore, metagenomic sequencing technology can be used to explore functional microorganisms with the potential of BAs’ synthesis and catabolism in *Hongqu* rice wine brewing, which is conducive to the subsequent abatement of BAs by regulating the core functional microorganisms.

Here, we aimed to investigate the effects of microbial community on the formation of BAs and volatile flavor components during the traditional brewing of *Hongqu* rice wine. Firstly, the composition and variation of BAs and volatile flavor components during *Hongqu* rice wine brewing were analyzed. Subsequently, metagenomic sequencing was used to analyze the microbial community during *Hongqu* rice wine brewing at the species level. Besides, statistical analysis was performed to reveal the potential correlations between dominant microbial species and BAs, as well as characteristic volatile components. Finally, the microbial enzymes and the functional microorganisms involved in BAs’ metabolism and characteristic volatiles’ formation in *Hongqu* rice wine brewing were excavated based on bioinformatical analysis. This study provides scientific insights into the metabolic mechanisms of BAs and flavor during *Hongqu* rice wine brewing and also provides directions for subsequent regulation of the flavor quality of *Hongqu* rice wine.

## 2. Materials and Methods

### 2.1. Hongqu Rice Wine Brewing and Sample Collection

In this study, two representative fermentation starters (*Hongqu*) were obtained from Fujian Baishuiyang Wine Co., Ltd. (Ningde, China) and Fujian Gutian Chengjiu Hongqu Co., Ltd. (Ningde, China), respectively. *Hongqu* rice wine was brewed with glutinous rice as the main raw material and *Hongqu* as the fermentation starter through the traditional brewing technique as follows: (1) Preliminary preparation: sterilize the water tank with boiling water at 100 °C; after the glutinous rice is washed, it is soaked in a sterilized tank for 12 h, then steamed at 100 °C for 45 min, and the steamed glutinous rice is cooled to room temperature. (2) Fermentation operation: the steamed glutinous rice, cold water, and wine starter (*Hongqu*) are put into the wine jar at a ratio of 10:15:1; the jar is wrapped with 8 layers of gauze and fermented in a constant greenhouse at 18 °C for 10 days; after the 10th day, the 8 layers of gauze are replaced with sterile plastic bags for sealed anaerobic fermentation for 35 days. 

Fermented samples were collected from three parallel fermentation tanks at ten time points during the brewing process (Days 1, 2, 3, 5, 7, 10, 15, 20, 30, 45), sealed in sterile sample vials for further testing of alcohol, reducing sugar, acidity, volatile profiles, BAs, and microbial community. In detail, from the fermented sample, 10 g was taken, put into 90 mL of sterile saline, shaken with a vortex shaker for 30 s, filtered through sterile gauze, and centrifuged at 8000 r/min for 10 min, and the supernatant was collected and stored at −20 °C for subsequent physicochemical analysis. Meanwhile, the obtained microbial pellets were used for propidium monoazide (PMA) treatment [20] and immediately stored at −80 °C for shotgun metagenomic sequencing analysis.

### 2.2. Determination of Total Acid Content

The total acid content of the wine mash in the *Hongqu* rice wine brewing was measured by a potentiometric titrimeter. In short, 5 mL of the supernatant of the *Hongqu* rice wine, 25 mL of CO_2_-free water, and a magnetic stirring rod were mixed in a 100 mL beaker. Then, the beaker was placed on an electromagnetic stirrer under agitation and titrated with NaOH standard solution (10 mM), with pH 8.20 as the titration end point.

### 2.3. Determination of Reducing Sugar Content

The 3,5-dinitrosalicylic acid (DNS) method [21] was used to determine the content of reducing sugar in the wine mash during the fermentation of the *Hongqu* rice wine. Briefly, 1 mL of moderately diluted sample was mixed with 2.0 mL DNS reagent and bathed in boiling water for 5 min. After that, it was quickly cooled with running water, supplemented with distilled water to 25 mL, shaken well, and finally, measured at 540 nm for absorbance and reducing sugar content according to the glucose standard curve.

### 2.4. Determination of Alcohol Content

Alcohol content was determined by a gas chromatograph (GC 7890A, Agilent, Palo Alto, CA, USA) with an FID detector and an Agilent HP-INNOWAX column (30 m length, 0.25 mm inner diameter, and 0.25 μm film thickness). The injection volume of the fermented sample was 1 μL, and the oven temperature was programmed as follows: 40 °C lasting for 5 min, then increasing to 220 °C at a lifting speed of 20 °C/min, and then holding at 220 °C for 5 min. The shunt ratio was 10:1. The flow rate of H_2_ was 30 mL/min. The flow rate of air was 400 mL/min. The flow rate of tail blow (N_2_) was 25 mL/min.

### 2.5. Determination of Biogenic Amine Content

Biogenic amines were extracted as described by Świder et al. [22]. The derivatization of the fermented samples was based on the Chinese standard GB 5009.208-2016 [23]. After filtration with a 0.22 μm nylon filter, samples after derivatization treatment were put into a 2 mL liquid injection bottle to be tested. The quantification of the BAs derivatives was carried out by using a HPLC instrumentation (LC-20A, Shimadzu, Tokyo, Japan) equipped with a UV detector (SPD-M20A) and a Supelcosil LC-18 column (4.6 × 250 mm, 5 μm). The HPLC mobile phase, as well as the detection parameters can be found in our previous study [11].

### 2.6. Volatile Organic Compound Analysis

The profile of volatile compounds was mainly determined using headspace solid-phase micro-extraction (HS-SPME) [24] combined with gas chromatography-mass spectrometry (GC-MS) (Agilent 7890-B/5977A MS system, Agilent Technologies, Little Falls, DE, USA). The sample treatment and instrument operating parameters were carried out according to the methods described in our previous study [25]. The chromatographic peaks were qualitatively determined by matching the NIST 17s mass spectral library and Kovats index, which was calculated by the analysis of the C7–C40 n-alkane standard (Sigma-Aldrich, St. Louis, MO, USA) under the same chromatographic conditions as the samples. The concentration of each volatile compound was calculated quantitatively by the normalized method based on the peak area of the internal standard compound (2-octanol, 10 mg/L) and the correction factor of the reference compounds purchased from Sigma-Aldrich Chemical Co. (St. Louis, MO, USA).

### 2.7. Microbial DNA Extraction and Metagenomic Analysis

Total microbial DNA was extracted from fermented samples (*Jiupei*) collected at Day 1, Day 3, Day 5, Day 10, Day 20, and Day 30 using the CTAB-based method [26]. Briefly, the mash sample (5 g) was mixed with 15 mL of DNA extraction buffer (100 mM Tris-HCl, 100 mM sodium EDTA, 100 mM sodium phosphate, 1.5 M NaCl, 1% CTAB, pH 8.0) and 100 mL of proteinase K (10 mg/mL) in a 50 mL Falcon tube with horizontal shaking at 200 rpm for 30 min at 37 °C. After shaking, 3 mL of 10% SDS was added, and the samples were incubated in a 65 °C water bath for 3 h with gentle end-over-end inversions every 15–20 min. The supernatants were collected after centrifugation at 6000× *g* for 10 min at room temperature and transferred into another 50 mL centrifuge tube. The pellets were extracted two more times by adding 4.5 mL of extraction buffer and 1 mL of 10% SDS, vortexing for 10 s, incubating at 65 °C for 10 min, and centrifuging as before. Supernatants from the three cycles of extraction were combined and mixed with an equal volume of chloroform-isoamyl alcohol (24:1, *v/v*). The aqueous phase was recovered by centrifugation and precipitated with 0.6 vol of isopropanol at room temperature for 1 h. A pellet of crude nucleic acids was obtained by centrifugation at 12,000× *g* for 30 min at room temperature, washed with pre-chilled 70% ethanol, and resuspended in sterile Tris-EDTA buffer (pH 8.0) to give a final volume of 500 mL. The concentration and purity of the extracted DNA were checked by 1.0% agarose gel electrophoresis under ultraviolet light and Thermo Qubit 3.0 (Thermo Fisher Scientific, Wilmington, NC, USA) and then stored at −80 °C for further metagenomic sequencing analysis. 

Metagenomic sequencing and annotation were supplied in the previous publication [27]. DNA library preparation followed the manufacturer’s instructions (Illumina). Paired-end sequencing, sequence statistics, and quality control were performed on Illumina NovaSeq Xten (Illumina Inc., San Diego, CA, USA) at Majorbio Bio-Pharm Technology Co., Ltd. (Shanghai, China) using NovaSeq Reagent Kits according to the manufacturer’s instructions [16]. Briefly, a total amount of 1 μg DNA per sample was used as the input material for the DNA sample preparations. Sequencing libraries were generated using the NEBNext^®^ Ultra™ DNA Library Prep Kit for Illumina (NEB, Ipswich, MA, USA) following the manufacturer’s recommendations, and index codes were added to attribute sequences to each sample. The base-calling pipeline (Version IlluminaPipeline-0.3) was used to process the raw fluorescent images and call sequences. The relative abundance of genus and functional species was examined using data from the NR database. High-quality short reads of each DNA sample were assembled by MetaVelvet [28]. BLASTx [29] was used to search the gene sequences of the predicted genes against GenBank’s non-redundant protein database (NR) and the Kyoto Encyclopedia of Genes and Genomes (KEGG) database with an e-value ≤ 1 × 10^−5^. The genes were annotated as a function of the NR or KEGG homologues with the lowest e-value. Genes that were annotated by KEGG were assigned into KEGG pathways. The functional genes and metabolic pathways associated with biogenic amines’ and characteristic volatile flavor components’ metabolism were investigated using the KEGG database (https://www.kegg.jp/, accessed on 1 October 2021). Metagenomic sequencing data have been uploaded to the NCBI Sequence Read Archive (SRA) database (accession number: PRJNA995035).

### 2.8. Multivariate Statistical Analysis and Visualization

The heatmap and bubble matrix package in the R software (Ver. 3.3.3) were used to visualize the volatile flavor profiles and the abundances of the functional genes encoding enzymes involved in BAs’ metabolism, respectively. SIMCA-14.1 software (Umetrics AB, Umea, Vasterbotten, Sweden) was used for principal component analysis (PCA) to characterize the dynamic changes of volatile flavor components during the *Hongqu* rice wine brewing. The microbial taxonomies with significant differences between HBAs and LBAs were revealed using STAMP software (Ver. 2.1.3) at the genus and species levels. The differences in the abundance of microbial taxonomies between HBAs and LBAs were determined using Welsh’s t-test, and the Benjamini–Hochberg procedure was used to control the false-discovery rate. Subsequently, the possible correlations between the key microbial species and BAs, as well as characteristic volatile components were calculated based on Spearman correlation coefficients and visualized through the R software (Ver. 3.3.3) with the Psych, Reshape2, and pheatmap packages.

## 3. Results and Discussion

### 3.1. The Dynamics of BAs during Hongqu Rice Wine Brewing

The most-common BAs in fermented foods include histamine, putrescine, cadaverine, tyramine, tryptamine, spermine, spermidine, 2-phenylethylamine, agmatine, etc. [30], among which the first seven BAs can be detected in the brewing process of *Hongqu* rice wines of HBAs and LBAs (Figure 1). The results of BAs content detection showed that the *Hongqu* rice wine fermented with the starter from Fujian Baishuiyang Wine Co., Ltd. (Pingnan, China), contained higher BAs (HBAs), while the *Hongqu* rice wine fermented with the starter from Fujian Chengjiu Wine Co., Ltd. (Gutian, China), contained lower BAs (LBAs). The BAs content of the *Hongqu* rice wine mainly accumulated in the early stage of fermentation (10 days of initial fermentation), followed by a slow increasing trend (Figure 1A). The main reason for the dramatic increase of the BAs’ accumulation in the early stage of the *Hongqu* rice wine brewing (Days 1–10) may be due to the decarboxylation of precursor amino acids by BAs-producing bacteria. The raw materials in the fermentation mash during this period were rich in nutrients, the microbial flora in the fermentation system was complex and diverse, and the microbial growth and metabolic activities were very vigorous. In particular, yeast had not produced a large amount of ethanol in the early stage of the *Hongqu* rice wine brewing, and there was a large number of BAs-producing bacteria with amino acid decarboxylase activity, which grew and metabolized during this period to produce large amounts of BAs [31]. At the end of fermentation, the total BAs content in the two types of *Hongqu* rice wine (HBAs and LBAs) was distinctly different (372.94 mg/L and 217.27 mg/L, respectively). In detail, tryptamine was the major BAs in both HBAs and LBAs, mainly accumulated in the early and middle stages of brewing, and the concentrations at the end of brewing were 123.01 mg/L and 176.46 mg/L, respectively (Figure 1B). In contrast, there was a significant difference in the content of putrescine in HBAs and LBAs, with the content of putrescine in HBAs being 129.15 mg/L, while the content of putrescine in LBAs was only 10.33 mg/L (Figure 1C). It is worth noting that cadaverine, histamine, and tyramine were only detected in the HBAs’ brewing process, of which cadaverine and tyramine mainly accumulated in large quantities within the first 10 days of fermentation (Figure 1D,F), while histamine showed a stable increasing trend throughout the HBAs’ brewing process (Figure 1E). Spermidine showed a trend of continuous accumulation during the brewing of HBAs, while its change was relatively stable during the brewing of LBAs (Figure 1G). The concentration of spermidine in both types of *Hongqu* rice wine was low. As for spermine, it increased continuously in both HBAs and LBAs. At the end of fermentation, the concentration of spermine was significantly different between HBAs and LBAs, and the content of spermine in LBAs was significantly higher (Figure 1H).

### 3.2. Physicochemical Parameters during Hongqu Rice Wine Brewing

Monitoring the changes in physicochemical parameters during *Hongqu* rice wine brewing may help to understand the living environment and metabolic activity of the microbial community. As shown in Figure 2A, the reducing sugar concentration of HBAs and LBAs increased sharply at the initial stage of fermentation and then declined rapidly, levelled off after 20 days, and reached 0.11 g/L and 0.29 g/L at the end of brewing, respectively. The sharp increase in reducing sugar content may be due to the hydrolysis of starch in raw materials to glucose under the action of α-amylase and glucoamylase from microorganisms such as *Monascus* spp. [32,33,34]. Reducing sugar is not only the main energy substance for microbial growth and reproduction, but also the carbon skeleton of substance metabolism. Microorganisms such as yeast and lactic acid bacteria can utilize reducing sugar to produce flavor substances including alcohols, esters, and organic acids, which may cause the continuous decline in reducing sugar concentration [35]. Accordingly, the alcohol and total acid contents increased rapidly in the early and middle stages of brewing for HBAs and LBAs and leveled off after 20 days of fermentation (Figure 2B,C). Among them, the alcohol content of HBAs and LBAs at the end of brewing was 17.84% (*v/v*) and 18.88% (*v/v*), respectively, with no significant difference (*p* > 0.05), while the total acid content was 9.77 g/L and 4.92 g/L, respectively, with a significant difference (*p* < 0.01), which may be related to the differences in the species and abundance of lactic acid bacteria in the two *Hongqu* rice wines [36].

### 3.3. Volatile Flavor Components during Hongqu Rice Wine Brewing

The composition and contents of volatile components are among the important indicators to evaluate the flavor quality of *Hongqu* rice wine. In this study, a total of 79 volatile compounds were detected and identified through SPME-GC-MS in the brewing samples of HBAs and LBAs, including esters (29), alcohols (21), acids (9), aldehydes (6), ketones (4), phenols (3), and others (7). As can be seen from the heatmap (Figure 3A), the majority of the volatile flavor components in the mash samples of HBAs and LBAs were relatively low at the initial stages of brewing. With the progress of brewing, the content of most volatile compounds in the mash samples of HBAs and LBAs gradually emerged and showed an increasing trend, but the composition and content of volatile compounds in HBAs and LBAs were obviously different. Esters and alcohols are the main volatile flavor components in *Hongqu* rice wine, which give a pleasant floral and fruit aroma [37], as well as a fullness and sweetness to the wine [38]. In this study, it was found that most esters and alcohols (including ethyl dl-2-hydroxycaproate (C27), diethyl succinate (C35), ethyl undecanoate (C70), ethyl caproate (C6), ethyl 2-phenylacetate (C43), 1-octanol (C28), isoamyl alcohol (C5), phenylethyl alcohol (C55), isobutyl alcohol (C3), 1-octen-3-ol (C19), 1-decanol (C41), and cedrol (C62)) accumulated in large quantities during the brewing process, especially in the middle and late stages of brewing, and this was more pronounced in LBAs (Figure 3A). In contrast, the synthesis of characteristic ethyl esters, such as ethyl linoleate (C76), ethyl myristate (C59), ethyl hydrocinnamate (C52), ethyl palmitate (C67), ethyl oleate (C73), ethyl lactate (C11), and ethyl heptanoate (C10), were more active in the brewing of HBAs relative to LBAs, which may be related to the higher abundance of *Saccharomyces* in the brewing process of HBAs and rapid growth to produce alcohols, because *Saccharomyces* is the main producer of flavor substances in alcoholic fermentation products, especially flavor substances such as alcohols and esters [35,39]. In addition, almost all volatile organic acids showed a steady increase during HBAs’ and LBAs’ brewing, which is similar to the result of our previous study [11]. Interestingly, phenolics (e.g., 2,4-di-t-butylphenol (C69), 2-methoxy-4-vinylphenol (C65)) and aldehydes (e.g., decanal (C23), benzaldehyde (C24), and nonanal (C16)) were detected in greater abundance in the traditional brewing of LBAs than in HBAs. Decanal, benzaldehyde, and nonanal were present at the beginning of LBAs’ fermentation, which may be derived from the wine starter. Four ketones, 2-nonanone (C15), geranylacetone (C48), gamma-nonalactone (C58), and acetophenone (C33), were detected in this study, among which 2-nonanone (C15) was the main ketone in HBAs, while acetophenone (C33) was the main ketone in LBAs. The above results compared the difference in the abundance of flavor components between HBAs and LBAs, and the main reason for these obvious differences may be the differences in the composition of the microbial communities during the brewing process of HBAs and LBAs, where different microbial flora may have certain effects on the production of esters, phenols, and other substances.

As shown in Figure 3B, the PCA score scatter plot shows that the dynamic characteristics of volatile components between LBAs and HBAs are obviously different. The fermented samples of LBAs and HBAs gradually separated as the brewing process proceeded, indicating that the differences in the composition and concentration of volatile flavor compounds between the two types of *Hongqu* rice wine gradually increased (Figure 3B). At the same time, the PCA loading scatter plot further supplements the characteristic volatile flavor components of the corresponding fermented samples (Figure 3C). The results showed that the volatile flavor profiles of HBAs and LBAs in the early stages of brewing were mainly characterized by L.alpha.-terpineol (C37), 2-chloroethanol (C14), 1-hexanol (C12), n-dibutylformamide (C40), nonaldehyde (C16), ethyl n-benzoylglycinate (C7), and ethyl caprylate (C18). In the later stages of brewing, the volatile flavor profiles of HBAs were mainly characterized by ethyl linoleate (C76), ethyl hydrocinnamate (C52), ethyl myristate (C59), ethyl lactate (C11), 2-nononone (C15), nonanoic acid (C64), 2-nonanol (C25), ethyl palmitate (C67), ethyl oleate (C73), and other substances that appear in the first quadrant, while those of LBAs were mainly characterized by coumaran (C71), 2-methoxy-4-vinylphenol (C65), isobutyl nonyl phthalate (C77), (3z)-3-nonen-1-ol (C36), 3-ethoxy-1-propanol (C13), and other substances appearing in the fourth quadrant. Finally, hierarchical cluster analysis was used to further reflect the differences in the characteristics of the volatile flavor components between the two kinds of *Hongqu* rice wine (LBAs and HBAs). As shown in Figure 3D, it can be seen from the hierarchical clustering diagram that the characteristics of the volatile components of LBAs and HBAs were similar at the initial stage of fermentation (the first 7 days), but after 7 days of fermentation, there were obvious differences in the volatile flavor characteristics between LBAs and HBAs, which can be clearly divided into two categories, Group 1 and Group 2 (Figure 3D). 

To further analyze the characteristics of the *Hongqu* rice wines of HBAs and LBAs in terms of volatile flavor components, we compared the inter-group differences in the concentration of volatile compounds in samples at the end of fermentation (Day 45) (Figure 4). Specifically, the alcohols significantly enriched in HBAs were 2-heptanol (C9), isopropyl alcohol (C30), and L.alpha.-terpineol (C37), whereas the alcohols significantly enriched in LBAs were isobutyl alcohol (C3), 3-ethoxy-1-propanol (C13), 1-octen-3-ol (C19), (3z)-3-nonen-1-ol (C36), 1-decanol (C41), and citronellol (C42). In addition, the contents of some esters in HBAs and LBAs were significantly different, with 13 and 15 esters significantly enriched in HBAs and LBAs, respectively. The esters significantly enriched in HBAs were ethyl lactate (C11), 3-methylbutyl methoxyacetate (C29), ethyl hydrocinnamate (C52), ethyl myristate (C59), ethyl palmitate (C67), ethyl n-octadecanoate (C72), ethyl oleate (C73), and ethyl linoleate (C76). The esters significantly enriched in LBAs were isopentyl acetate (C4), 4-hydroxybutyric acid (C31), ethyl caprate (C32), butyl ethyl succinate (C44), phenethyl acetate (C46), ethyl 3-methylbutyl succinate (C54), and isopropyl palmitate (C66). 

### 3.4. Microbial Communities during Hongqu Rice Wine Brewing

The traditional brewing of *Hongqu* rice wine is carried out in an open environment, involving the metabolic activities of various microorganisms. Revealing the differences in microbial community structure between HBAs and LBAs during brewing is helpful to explain the differences in the formation of volatile flavor components and biogenic amines between HBAs and LBAs. At the level of bacterial genera, the dominant bacteria in the brewing of HBAs were mainly *Pantoea*, *Klebsiella*, *Leuconostoc*, *Lactobacillus*, and *Lactococcus*, while the dominant bacteria in the brewing of LBAs were mainly *Weissella*, *Pediococcus*, *Leuconostoc*, *Lactobacillus*, and *Streptococcus* (Appendix A). Among them, *Leuconostoc*, *Lactobacillus*, *Lactococcus*, *Weissella*, and *Pediococcus* all belong to lactic acid bacteria, indicating that lactic acid bacteria may play an important role in the brewing process of *Hongqu* rice wine, which is similar to our previous findings [11]. It is well known that most of the lactic acid bacteria have good acid-producing properties, and the differences in lactic acid bacteria species and their abundance may be the main reason for the differences in total acid content between the two *Hongqu* rice wines (HBAs and LBAs). The result of the principal component analysis (PCA) showed that the bacterial community characteristics of HBAs and LBAs were obviously different, indicating that the bacterial composition and their relative abundance in the brewing of HBAs and LBAs differed significantly (Appendix A). Visual analysis using STAMP software revealed bacterial genera with significant differences between HBAs and LBAs, and the results showed that the abundance of bacteria such as *Klebsiella*, *Salmonella*, and *Serratia* was significantly higher in the brewing process of HBAs compared to LBAs, while the abundance of bacteria such as *Pediococcus*, *Enterococcus*, and *Bacillus* were significantly lower in the HBAs’ brewing process (Appendix A). As a highly acid-tolerant parthenogenic anaerobic bacterium, *Klebsiella* metabolizes primarily through respiration and fermentation, using citrate and glucose as the only carbon sources, fermenting glucose and producing acids and gases [27].

At the fungal genus level, *Monascus*, *Aspergillus*, and *Saccharomyces* were the three most-important fungi in the brewing of the *Hongqu* rice wine (Appendix A). Among them, *Monascus* was the main fungal genus during the fermentation of HBAs, and *Saccharomyces* dominated the fermentation of HBAs after the third day of fermentation. In the brewing process of LBAs, *Aspergillus* and *Saccharomyces* jointly dominated the fungal flora, and the relative abundance of *Monascus* was lower, but still occupied a certain abundance in the whole brewing process. According to research reports, as a characteristic microorganism in *Hongqu* fermentation, *Monascus* can metabolize and produce pigment, γ-aminobutyric acid, monacolin K, and other secondary metabolites with physiological activity [40,41,42]. The score plot of the principal component analysis (PCA) shows that the fungal community characteristics of all fermentation samples were relatively similar, except for the first day fermentation sample, indicating that the changes of the fungal community composition in the *Hongqu* rice wine brewing were relatively stable, but there were significant differences in the fungal community characteristics between HBAs and LBAs (Appendix A). The results of visual analysis based on the STAMP software showed that the abundance of *Candida* during the fermentation of HBAs was significantly higher than that of LBAs, while the abundance of *Aspergillus* was significantly lower than that of LBAs (Appendix A).

The results of the analysis at the species level revealed that the dominant bacteria in the brewing of HBAs were unclassified_g_*Pantoea*, *Klebsiella pneumoniae*, *Panobacter disperse*, unclassified_f_*Enterobacteriaceae*, and *Leuconostoc mesenteroides*, while the dominant bacteria during the fermentation of LBAs were *Weissella confuse* and *Pediococcus acidilactici* (Figure 5A). In terms of fungal species, *Monascus purpureus* was the main fungal species in the early stage of HBAs’ fermentation, and *Saccharomyces cerevisiae* became the absolute dominant fungus after 3 days of fermentation. *Aspergillus niger* and *Saccharomyces cerevisiae* were the two dominant fungi in the brewing process of LBAs, followed by *Monascus purpureus*, *Aspergillus welwitschiae*, *Aspergillus awamori*, and *Aspergillus phoenicis* (Figure 5B). The results of the PCA at the microbial species level were similar to those at the genus level, and both indicated significant differences in the microbial community composition between HBAs and LBAs (Figure 5C,D). The visualization results based on the STAMP software analysis showed that the relative abundances of *Klebsiella pneumoniae*, *Pantoea dispersa*, *Leuconostoc mesenteroides*, *Lactococcus lactis*, *Klebsiella quasipneumoniae*, *Klebsiella variicola*, *Klebsiella aerogenes*, *Escherichia coli*, and *Saccharomyces boulardii* (nom. inval.) in HBAs’ brewing were significantly higher than those in LBAs, while the relative abundances of *Weissella confusa*, *Pediococcus acidilactici*, *Pediococcus pentosaceus*, *Lactobacillus brevis*, *Streptococcus mutans*, *Weissella cibaria*, *Streptococcus pneumoniae*, *Streptococcus salivarius*, *Aspergillus niger*, *Aspergillus phoenicis*, *Aspergillus awamori*, *Aspergillus welwitschiae*, *Aspergillus luchuensis*, *Aspergillus eucalypticola*, and *Aspergillus vadensis* in LBAs were significantly higher than those in HBAs (Figure 5E,F).

### 3.5. Correlation Analysis of Predominant Microbial Phylotypes and Characteristic Metabolites in Hongqu Rice Wine Brewing

To a large extent, the formation of flavor components and potential hazards in *Hongqu* rice wine brewing depends on the composition of the microbial community. Therefore, it is necessary to elucidate the potential correlation between the key microbial phylotypes and characteristic metabolites in the fermentation of *Hongqu* rice wine, in order to provide a theoretical basis for controlling the flavor quality and drinking safety of *Hongqu* rice wine. In this study, Spearman’s correlation between the predominant microbial species (top 60 fungi and bacteria) and the characteristic volatile components, as well as BAs are visualized through a correlation heatmap (Figure 6). The results showed that putrescine, tyramine, histamine, cadaverine, tryptamine, spermidine, spermine, and total BAs were positively correlated with *Lactobacillus curvatus*, *Lactococcus lactis*, and *Leuconostoc mesenteroides.* In addition, putrescine, tyramine, histamine, and cadaverine had highly similar associations with the microbiota, showing significant positive associations with *Salmonella enterica*, unclassified_f_*Enterobacteriacea*, and most *Klebsiella* (including *Klebsiella variicola*, *Klebsiella aerogenes*, *Klebsiella quasipneumoniae*, and *Klebsiella pneumoniae*), but negatively correlated with *Pediococcus acidilactici*, *Sphingomonas* spp., *Kosakonia cowanii*, and *Klebsiella* cf. *planticola* B43 (Figure 6A). Previous studies have shown that the production of biogenic amines in fermented food is closely related to the metabolic activities of bacteria [43,44]. In addition, the production of BAs is associated with the protective effect of bacteria against the acidic fermentation environment. For example, lactic acid bacteria produce biogenic amines during the fermentation of foods to maintain their survival in an acidic environment [45]. 

Compared to the bacterial community, with the exception of *Saccharomyces*, the fungal community showed less-positive correlations with the characteristic volatile components, as well as biogenic amines (Figure 6B). Almost all biogenic amines were positively correlated with *Saccharomyces* (including *Saccharomyces uvarum*, *Saccharomyces boulardii*, *S. cerevisiae* × *S. kudriavzevii*, *Saccharomyces paradoxus*, *Saccharomyces cerevisiae*, and *Saccharomyces pastorianus*), while negatively correlated with *Aspergillus* (such as *Aspergillus piperis*, *Aspergillus luchuensis*, *Aspergillus tubingensis*, *Aspergillus niger*, *Aspergillus parasiticus*, *Aspergillus oryzae*, etc.) and *Monascus* (*Monascus ruber*, *Monascus purpureus*, *Monascus pilosus*). A previous study found that yeasts including *Saccharomyces* also have the ability to produce biogenic amines. In addition, all *Saccharomyces* species showed a strong positive correlation with alcohols and esters, including ethyl n-octadecanoate, 1-decanol, ethyl n-octadecanoate, citronellol, butyl ethyl succinate, ethyl 3-methylbutyl succinate, ethyl oleate, ethyl palmitate, 3-methylbutyl methoxyacetate, ethyl hydrocinnamate, ethyl lactate, ethyl myristate, ethyl linoleate, isopropylalcohol, and 2-heptanol [46,47]. In addition, putrescine, tyramine, histamine, spermidine, and cadaverine had negative correlations with most dominant fungi, including *Aspergillus*, *Rasamsonia*, *Talaromyces*, *Penicillium*, *Monascus*, and *Byssochlamys*. *Aspergillus niger*, *Aspergillus luchuensis*, *Aspergillus neoniger*, and *Aspergillus sclerotiicarbonarius* showed strong positive correlations with L.alpha.-terpineol, (3Z)-3-nonen-1-ol, isobutyl alcohol, 3-ethoxy-1-propanol, 1-octen-3-ol, and cedrol [25]. 

### 3.6. Functional Genes and Microorganisms Related to the Metabolism of Biogenic Amines and Characteristic Flavor Components

Combining the KEGG database (https://www.kegg.jp/kegg/, accessed on 1 October 2021) and relevant literature [9,16,17], we searched for enzymes responsible for the metabolism of BAs and characteristic flavor components and mapped the bubble plot of the corresponding functional genes encoding these enzymes during the brewing of HBAs and LBAs (as shown in Figure 7). The annotated information of functional-gene-related microbial species is shown in Appendix A. BAs are mainly produced by the conversion of amino acids under the catalysis of the corresponding decarboxylase [48]. As the main precursors of biogenic amines, the efficiency of amino acids’ metabolism affects biogenic amines’ formation. In this study, the abundances of genes involved in histidine, lysine, and ornithine synthesis in HBAs’ brewing were much higher than those in LBAs’ brewing, possibly contributing more amino acid precursors during the brewing of HBAs (Figure 7A). Meanwhile, the encoding genes for histidine decarboxylase (EC 4.1.1.22, catalyzing histidine to histamine), ornithine decarboxylase (EC 4.1.1.17, catalyzing ornithine to putrescine) [49], D-ornithine/D-lysine decarboxylase (EC 4.1.1.116, catalyzing ornithine to putrescine) [50], and lysine decarboxylase (EC 4.1.1.18, catalyzing lysine to cadaverine) [51] were also obviously higher during HBAs’ brewing. The synergistic interaction between precursor amino acid synthesis and decarboxylation may be the main reason for the high levels of histamine, cadaverine, and putrescine in the brewing of HBAs. *Pantoea dispersa*, *Klebsiella pneumoniae*, unclassified_g_*Pantoea*, unclassified_f*_Enterobacteriaceae*, and *Saccharomyces cerevisiae* were identified to be the main microbial species responsible for amino-acid-decarboxylase-encoding genes (Appendix A), which were closely related to the synthesis of histamine, cadaverine, and putrescine in the *Hongqu* rice wine brewing. In addition, putrescine can also be further derived from the conversion of agamidine produced by arginine decarboxylation, which mainly involves arginine decarboxylase (EC 4.1.1.19, catalyzing arginine to agmatine) [52], agmatinase (EC 3.5.3.11, catalyzing agmatine to agmatine), and agmatine deiminase (EC 3.5.3.12, catalyzing agmatine to N-carbamoyl putrescine) [53]. *Klebsiella pneumoniae*, unclassified_g_*Pantoea*, *Lactococcus lactis*, and *Lactobacillus brevis* were the major contributors of arginine decarboxylase (EC 4.1.1.19), agmatinase (EC 3.5.3.11), and agmatine deiminase (EC 3.5.3.12). Subsequently, putrescine can also be converted to spermidine and spermine in a pathway involving spermidine synthase (EC 2.5.1.16, catalyzing arginine to spermidine) [54] and glutathionylspermidine amidase (EC 3.5.1.78). In this study, the abundance of genes encoding spermidine synthase (EC 2.5.1.16) and glutathione spermidine amidase (EC 3.5.1.78) during HBAs’ brewing was obviously higher than those of LBAs, which may be the main reason for the higher spermidine content in HBAs’ brewing (Figure 7A). The functional microorganisms for spermidine synthesis were mainly *Pantoea dispersa*, *Klebsiella variicola*, and *Klebsiella quasipneumoniae* in HBAs’ brewing. Conversely, the encoding gene abundance of spermine synthase (EC 2.5.1.22) was higher in LBAs’ brewing, but not in HBAs’ brewing, which could explain the higher spermine content in LBAs’ brewing (Appendix A). The encoding gene for spermine synthase (EC 2.5.1.22) was entirely derived from *Saccharomyces cerevisiae* in both HBAs’ and LBAs’ brewing. The content of tryptamine was high in both HBAs and LBAs, the synthesis of which mainly involved L-tryptophan decarboxylase (EC 4.1.1.105) [55] and aromatic-L-amino acid decarboxylase (EC 4.1.1.28) [56]. In this study, the microbial annotation results of the functional genes showed that *Monascus purpureus* was the main contributor to tryptamine biosynthesis genes in HBAs’ brewing, while *Aspergillus niger* was the main contributor to tryptamine biosynthesis genes in LBAs’ brewing, followed by *Monascus purpureus*.

Our previous study showed that lactic acid and acetic acid were the major organic acids in the alcoholic fermentation of *Hongqu* rice wine [57]. The abundance of most genes related to lactic acid and acetic acid metabolism was higher in HBAs than in LBAs during brewing (Figure 7B), which may be responsible for the higher total acid content in HBAs. Among them, the genes involved in lactic acid synthesis, such as L-lactate dehydrogenase (EC 1.1.1.27, catalyzing the interconversion of L-lactic acid and pyruvate), D-lactate dehydrogenase (EC 1.1.1.28, catalyzing the interconversion of D-lactic acid and pyruvate), and lactaldehyde dehydrogenase (EC 1.2.1.22, catalyzing lactaldehyde to lactic acid) were mainly contributed by *Leuconostoc mesenteroides*, *Lactobacillus plantarum*, *Klebsiella pneumoniae*, *Pediococcus acidilactici*, *Weissella confusa*, and *Saccharomyces cerevisiae* (Appendix A). Alcohols, aldehydes, carboxylates, and esters were the main volatile flavor compounds in the *Hongqu* rice wine. Among them, *Saccharomyces cerevisiae*, *Aspergillus niger*, *Monascus purpureus*, and unclassified_g_*Pantoea* were the major microbial contributors of alcohol dehydrogenase (EC 1.1.1.1, catalyzing the interconversion of alcohol and aldehyde or ketone), aldehyde dehydrogenase (NAD+) (EC 1.2.1.5, catalyzing the interconversion of aldehyde and carboxylate), and aldehyde dehydrogenase (NAD(P)+) (EC 1.2.1.3, catalyzing the interconversion of aldehyde and carboxylate) in the *Hongqu* rice wine brewing (Appendix A). The microbial synthesis of esters is mainly carried out by reversible catalysis of esterases and alcohol acyltransferases [58]. *Saccharomyces cerevisiae* was annotated as the main species contributor of alcohol-O-acetyltransferase-encoding genes (EC 2.3.1.84, catalyzing the interconversion of alcohol and acetyl ester) in the *Hongqu* wine brewing, closely related to the formation of ethyl esters. On the other hand, the microbial contributors to encoding genes of acetyl esterase (EC 3.1.1.1, catalyzing the interconversion of ester to alcohol and acid) were mainly *Lactobacillus curvatus*, *Lactobacillus plantarum*, and *Aspergillus niger*. Meanwhile, there were obvious phenolic concentration differences between the HBAs’ and LBAs’ brewing. Correspondingly, the abundance of genes related to phenolic metabolism in LBAs’ brewing was much higher than that in HBAs’ brewing (Figure 7B), possibly contributing more synthetic precursors to phenols such as 2-methoxy-4-vinylphenol and 2,4-di-t-butylphenol. The microbial annotation results showed that *Saccharomyces cerevisiae*, *Monascus purpureus*, and *Aspergillus* spp. were the main microbial contributors of phenolic-metabolism-related functional genes in the *Hongqu* rice wine brewing.

## 4. Conclusions

Exploring the influence of microbial communities on the formation of BAs and volatile flavor components in *Hongqu* rice wine production is an important prerequisite for improving the flavor quality. Histamine, putrescine, cadaverine, tyramine, tryptamine, spermine, and spermidine were detected to be the main BAs in the *Hongqu* rice wine. Metagenomic sequencing was conducted to profile the predominant microbiota during the *Hongqu* rice wine fermentation, and results showed that BAs’ metabolism involved a variety of enzymes and microorganisms. Statistical correlation analysis demonstrated that BAs were positively related to *Lactobacillus curvatus*, *Lactococcus lactis*, and *Leuconostoc mesenteroides*. Bioinformatical analysis revealed that the microbial genes encoding enzymes involved in BAs’ synthesis were more abundant in HBAs, and the microbial genes encoding enzymes related to BAs’ degradation and characteristic volatile components were more abundant in LBAs. This study provided important scientific data to elucidate the effects of microbial communities on the formation of volatile compounds and BAs in the *Hongqu* rice wine brewing. However, the bioinformatics analysis is only a predictive study result, and more experiments are needed to further validate it. In a future study, it will be necessary to elucidate the formation and regulatory mechanisms of BAs and volatile compounds in *Hongqu* rice wine brewing through biofortification to provide technical support for the healthy development of the *Hongqu* rice wine industry.

## Figures and Tables

**Figure 1 foods-12-03075-f001:**
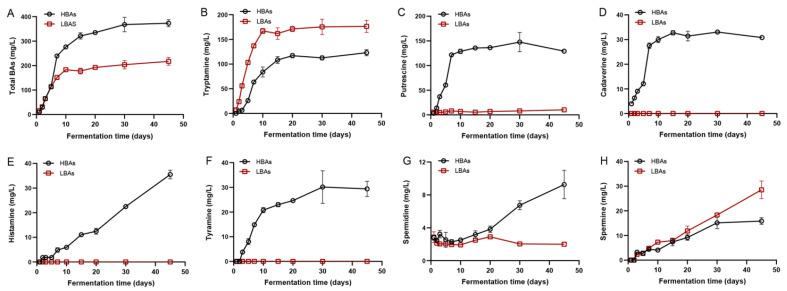
The dynamic change of biogenic amines (BAs) during the traditional brewing of *Hongqu* rice wine. (**A**) Total BAs, (**B**) tryptamine, (**C**) putrescine, (**D**) cadaverine, (**E**) histamine, (**F**) tyramine, (**G**) spermidine, and (**H**) spermine. HBAs: *Hongqu* rice wine with high BAs content; LBAs: *Hongqu* rice wine with low BAs content.

**Figure 2 foods-12-03075-f002:**
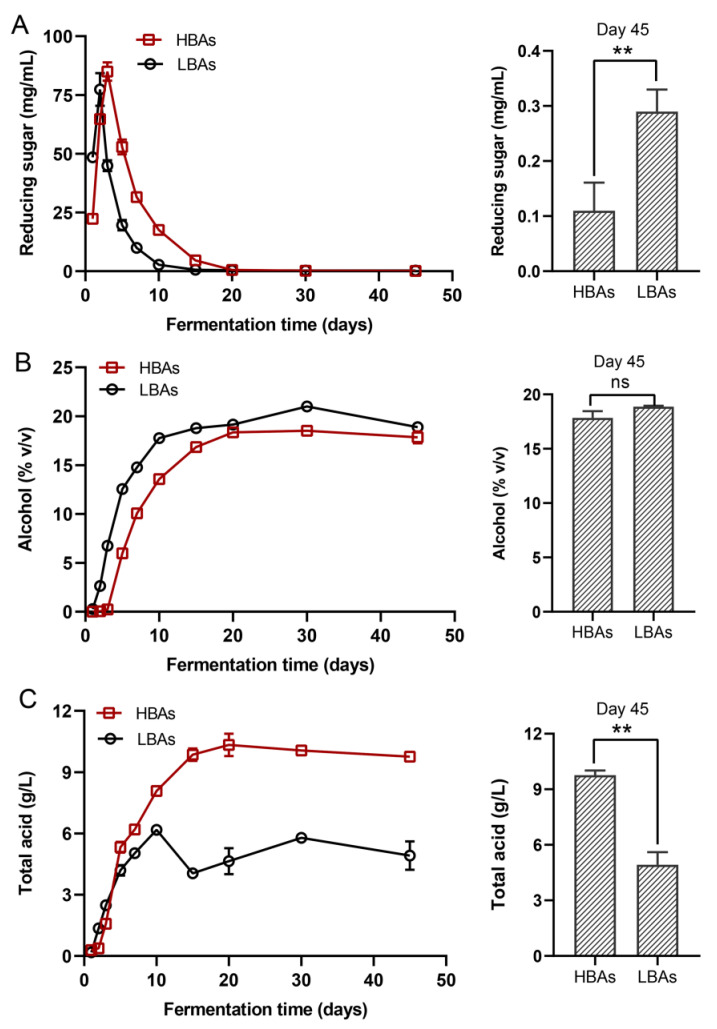
The changes in the physicochemical parameters of *Hongqu* rice wine fermentation. (**A**) Reducing sugar, (**B**) alcohol, (**C**) total acids. Student’s t-test was used for two-group comparisons (** *p* < 0.01).

**Figure 3 foods-12-03075-f003:**
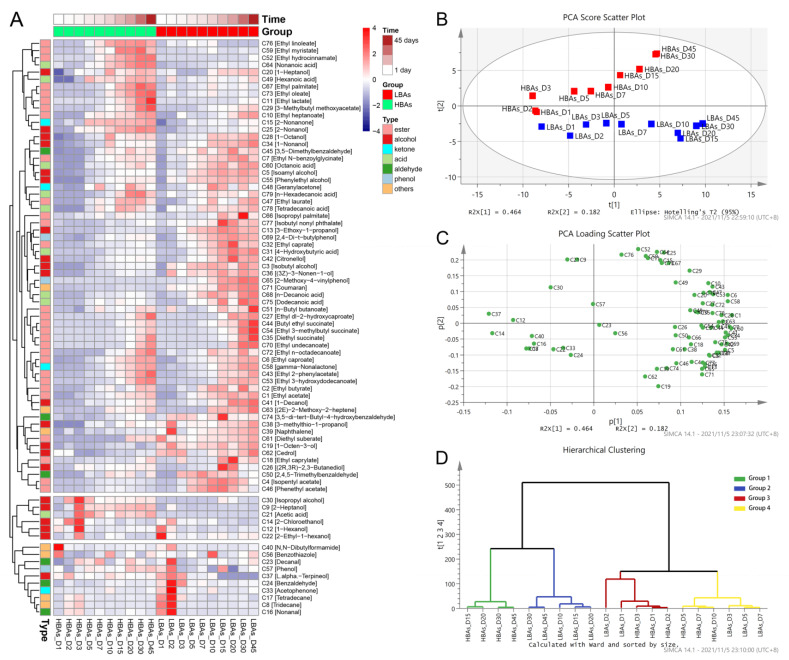
The dynamic changes of volatile flavor components during the traditional brewing of Hongqu rice wine. (**A**) Heatmap analysis of volatile components during the traditional brewing of HBAs and LBAs. (**B**) Principal component analysis (PCA) score scatter plot. (**C**) PCA loading scatter plot. (**D**) Hierarchical clustering diagram.

**Figure 4 foods-12-03075-f004:**
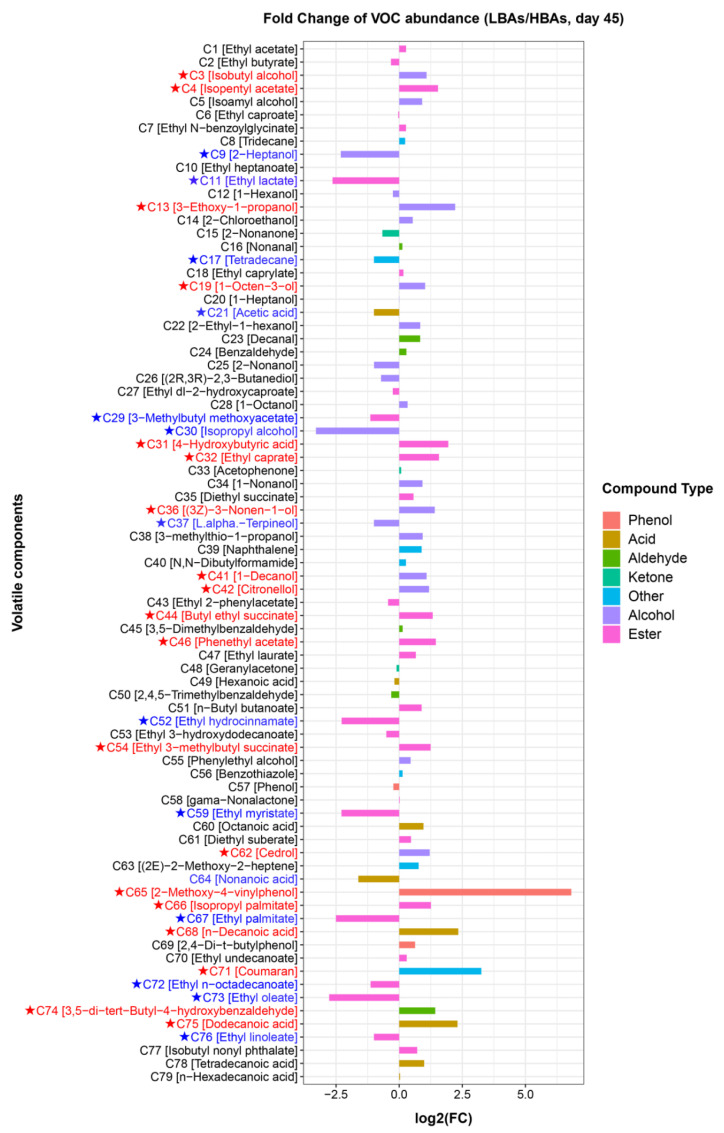
Comparison of the abundances of volatile flavor components in LBAs and HBAs through fold change chart. A log2 (LBAs/HBAs) value greater than 1 or less than −1 combined with a *p*-value of less than 0.05 for one-way ANOVA indicates a significant difference in the abundance of this volatile substance. Volatile compounds labeled in red with a log2(LBAs/HBAs) value greater than 1 indicate that these volatile compounds are enriched in LBAs; volatile compounds labeled in blue with a log2(LBAs/HBAs) value less than −1 indicate that these volatile compounds are enriched in HBAs.

**Figure 5 foods-12-03075-f005:**
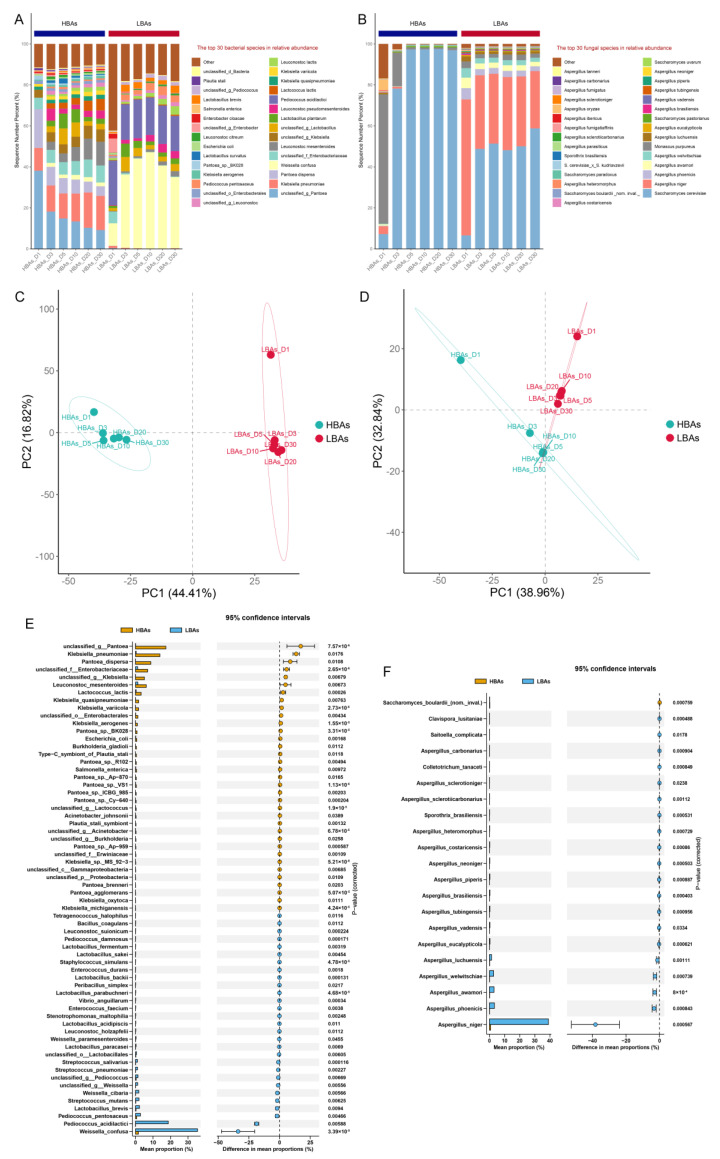
Analysis of microbial species in HBAs’ and LBAs’ brewing. The relative abundance of the predominant bacterial (**A**) and fungal (**B**) species. Principal component analysis (PCA) of bacterial (**C**) and fungal (**D**) communities. Visualization of the differences in the relative abundance of bacterial (**E**) and fungal (**F**) species between HBAs and LBAs. Microbial species with significant differences between HBAs and LBAs were determined using Welsh’s t-test, and the Benjamini–Hochberg procedure was used to control the false-discovery rate due to multiple testing. Corrected *p*-values are shown on the right.

**Figure 6 foods-12-03075-f006:**
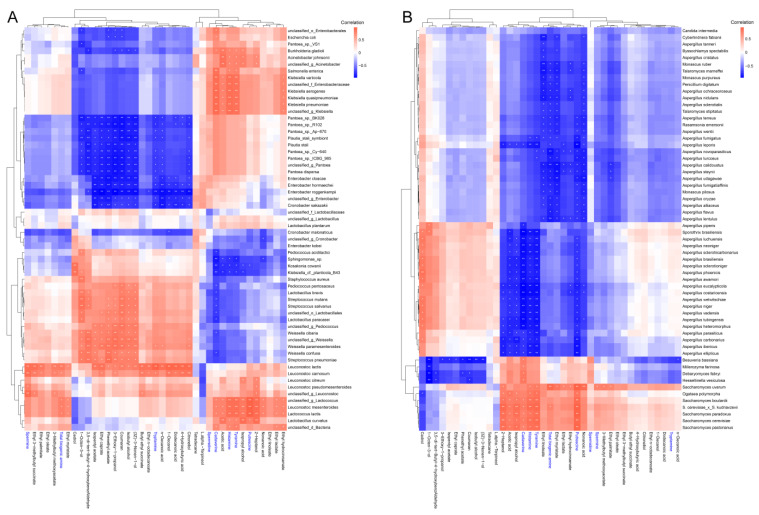
Correlation analysis between the characteristic volatile components, biogenic amines (BAs), and the predominant microbial phylotypes at the species level during the brewing of *Hongqu* rice wine. (**A**) Bacteria: characteristic volatile components and BAs; (**B**) fungi: characteristic volatile components and BAs. (0.01 < * *p* < 0.05, ** *p* < 0.01).

**Figure 7 foods-12-03075-f007:**
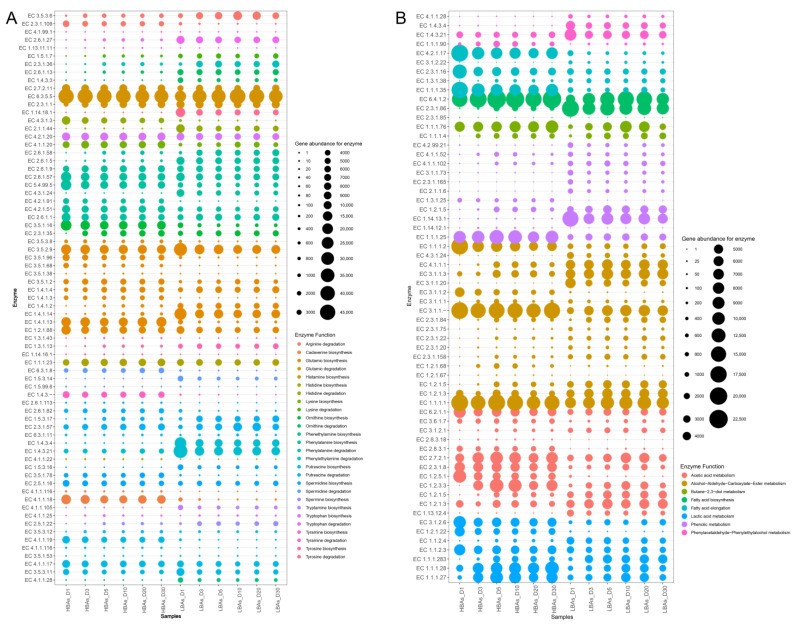
Bubble chart of the abundance of microbial enzymes closely related to the metabolism of biogenic amines (**A**) and characteristic flavor components (**B**) in HBAs and LBAs.

## Data Availability

The data presented in this study are available on request from the corresponding author.

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
