# Peer review of "Metagenomic Insights into the Regulatory Effects of Microbial Community on the Formation of Biogenic Amines and Volatile Flavor Components during the Brewing of Hongqu Rice Wine"

_foods, 2023, doi:10.3390/foods12163075_

Round 1

Reviewer 1 Report

The work described in this manuscript is very interesting and relevant due to the importance of rice wine and the impact of biogenic amines in health. In my opinion, the manuscript is original. It is scientifically well. The data are well discussed. My concern with this manuscript is the presentation. I found some minor details in this regard. For example, the authors did not explain the meaning of LBAs and HBAs in the abstract. I also found repeated words (e.g., putrescine and putrescine in page 2) and other minor details. I suggest reviewing the presentation of the paper.

Author Response

Q1: My concern with this manuscript is the presentation. I found some minor details in this regard. For example, the authors did not explain the meaning of LBAs and HBAs in the abstract. I also found repeated words (e.g., putrescine and putrescine in page 2) and other minor details. I suggest reviewing the presentation of the paper.

R1: Thanks for your valuable advice. The explanation of the full names of HBAs and LBAs in the Abstract does allow readers to more quickly understand the research framework of the manuscript. In addition, the manuscript we submitted before did have some mistakes in the description of repeated words. We have revised the manuscript in accordance with your suggestions. For details, please refer to the revised manuscript we submitted.

Reviewer 2 Report

The purpose of this research was to examine the role that the microbiome plays in controlling the development of BAs and volatile flavor components during the production of Hongqu rice wine. Histamine, putrescine, cadaverine, tyramine, tryptamine, spermine, and spermidine were found to be the most abundant BAs in Hongqu rice wine. Hongqu rice wine of HBAs had considerably greater concentrations of putrescine, cadaverine, histamine, tyramine, and spermidine than LBAs' did. The study also found that the abundances of microbial genes encoding enzymes associated to BAs breakdown and the metabolism of distinctive volatile components were greater in LBAs than they were in HBAs. Overall, this effort gives valuable scientific data for improving the taste of Hongqu rice wine and paves the way for the sector to flourish.

The research is intriguing, however it needs to be revised to increase its readability and quality.

Section 3.1

-While it was stated that the BAs content of Hongqu rice wine primarily accumulates in the initial 10 days of fermentation, followed by a gradual increase trend, the authors can substantiate this by explaining why these observations were made.

Section 3.2

- Esters, phenolics, and alcohol were found to vary considerably during the brewing process, particularly in the middle and late stages. The authors can expand upon their discussion of these observations.

General:

-This work's novelty should be highlighted and emphasized. Currently, it does not really stand out.

-Authors should proofread their work for errors in grammar and spelling.

Authors should proofread their work for errors in grammar and spelling.

Author Response

Q1: Section 3.1: While it was stated that the BAs content of Hongqu rice wine primarily accumulates in the initial 10 days of fermentation, followed by a gradual increase trend, the authors can substantiate this by explaining why these observations were made.

R1: Thanks for your valuable suggestions. According to your suggestions, we have made corresponding supplements in Section 3.1 of the revised manuscript. “The main reason for the dramatic increase of BAs accumulation in the early stage of Hongqu rice wine brewing (day 1-10) may be due to decarboxylation of precursor amino acids by BAs-producing bacteria. The raw materials in the fermentation mash during this period are rich in nutrients, the microbial flora in the fermentation system is complex and diverse, and the microbial growth and metabolic activities are very vigorous. In particular, yeast has not produced a large amount of ethanol in the early stage of Hongqu rice wine brewing, and there are a large number of BAs-producing bacteria with amino acid decarboxylase activity, which grow and metabolize during this period to produce large amounts of BAs [30]." 

Reference:

[30] Xia, X.; Luo, Y.; Zhang, Q.; et al. Mixed Starter Culture Regulates Biogenic Amines Formation via Decarboxylation and Transamination during Chinese Rice Wine Fermentation. J. Agric. Food Chem. 2018, 66 (25), 6348-6356.

Q2: Section 3.3: Esters, phenolics, and alcohol were found to vary considerably during the brewing process, particularly in the middle and late stages. The authors can expand upon their discussion of these observations.

R2: Thanks for your valuable suggestions. According to your comments, we have made corresponding additions and improvements to the manuscript, and your comments have greatly helped to improve the quality of our manuscript. For details, please refer to the revised manuscript we submitted.

Section 3.3: “......In contrast, the synthesis of characteristic ethyl esters, such as ethyl linoleate [C76], ethyl myristate [C59], ethyl hydrocinnamate [C52], ethyl palmitate [C67], ethyl oleate [C73], ethyl lactate [C11] and ethyl heptanoate [C10], were more active in the brewing of HBAs relative to LBAs, which may be related to the higher abundance of Saccharomyces in the brewing process of HBAs and rapid growth to produce alcohols, because Saccharomyces is the main producer of flavor substances in alcoholic fermentation products, especially alcohol and ester flavor substances [34,38].”

Section 3.3: “......The above results compared the difference in the abundance of flavor components between HBAs and LBAs, and the main reason for these obvious differences may be the differences in the composition of the microbial communities during the brewing process of HBAs and LBAs, where different microbial flora may have certain effects on the production of esters, phenols, and other substances.”

Reviewer 3 Report

Yang et al. investigated the regulatory effects of the microbial community on the formation of biogenic amines and volatile flavor components during the brewing of Hongqu rice wine. A lot of experiments were done and provided valuable results. However, a few points need clarification and improvement in the manuscript. Here are some comments on this paper and questions that I would like to discuss with the authors: 

1.        In the manuscript the line numbers are missing, it would be hard to point out the questions.

2.        I was really confused about what were HBAs and LBAs until I saw result 3.1. HBAs and LBAs should be described in the abstract.

3.        In this study, it appeared that the fermentation starter determined the outcome of the brew, so in addition to sequencing the brewing process, I would suggest sequencing the fermentation starter. This allowed to find out the causes of the increase in BAs.

4.        Information about metagenomic analysis is missing, I would like the authors could add detailed information about the processing.

5.        In section 3.1, “(HBAs and LBAs) were distinctly different (372.94 mg/L and 217.27 mg/L, respectively)” Could authors provide any explanation why the concentrations of BAs were significantly higher than 80 mg/L mentioned in the introduction?

6.        The results of volatile flavor components were tested and analyzed for 45 days, while the results of metagenomic changes were tested only for 30 days. Did the authors analyze the correlation between metabolism and microbiome on Day 30?

7.        I would suggest showing only the top 10 or 15 bacteria genres or species relative abundance in Figures 5 and 6. Only keep a few bacteria or fungi with high relative abundance in Figures 5 and 6 E and F.

8.        The metagenomic sequencing method was used in this study, as the authors described in the introduction its features, I would recommend that the genus analysis results could be placed in the supplementary file.

9.        In this study, time series data were obtained and it was possible to analyze the relationship between changes in the group's microbiomes and BAs.

10.     Statistical information is required in the figure legend, as in Figures 1, 2, and 4.

11.     Could authors upload the raw sequencing data to NCBI or other platforms? Because public sharing of sequencing data is an important part of microbiome research.

Author Response

Q1: In the manuscript the line numbers are missing, it would be hard to point out the questions.

R1: We apologize for the inconvenience caused by the missing of line numbers in the previously submitted manuscript. We have re-edited the line numbers in the revised manuscript for your further review.

Q2: I was really confused about what were HBAs and LBAs until I saw result 3.1. HBAs and LBAs should be described in the abstract.

R2: We apologize for not mentioning the exact meaning or definition of "HBAs and LBAs" in the Abstract and Introduction, which would have caused great inconvenience to the readers in reading the article as they could not quickly access the main research framework of the article. Following your suggestion, we have clarified the definitions of HBAs and LBAs in the abstract of the revised manuscript. The specific content of the modification is as follows:

Abstract: “The contents of putrescine, cadaverine, histamine, tyramine and spermidine in Hongqu rice wine of HBAs (with higher BAs content) were significantly higher than those in LBAs (with lower BAs content). ”

Q3: In this study, it appeared that the fermentation starter determined the outcome of the brew, so in addition to sequencing the brewing process, I would suggest sequencing the fermentation starter. This allowed to find out the causes of the increase in BAs.

R3: Thanks for your suggestion. In fact, we have conducted a lot of research on the microbial community of fermentation starter used in the brewing of Hongqu rice wine in our previous studies. The microorganisms in fermentation starter indeed play a crucial role in the formation of the flavor quality of Hongqu rice wine. However, the influence of microorganisms in fermentation starter on the formation of flavor and quality of Hongqu rice wine is mainly through the complex metabolism of microbial communities at different stages of the brewing process. It is still difficult to predict the flavor quality of rice wine by analyzing only the microbial composition of fermentation starter, because the microbial community during the brewing process of rice wine is in a state of succession and change. Only by analyzing the composition of microbiota and the variation trend of functional genes in brewing process can we understand the formation of flavor quality of rice wine more deeply.

Q4: Information about metagenomic analysis is missing, I would like the authors could add detailed information about the processing.

R4: Thanks for your suggestion. We are sorry that the description of metagenomic analysis information was not detailed enough in the manuscript submitted before to present more comprehensive information to readers. We have refined the content of metagenomic analysis and added information about metagenomic analysis in the revised manuscript. Thank you very much for providing us with such meaningful suggestion.

Q5: In section 3.1, “(HBAs and LBAs) were distinctly different (372.94 mg/L and 217.27 mg/L, respectively)” Could authors provide any explanation why the concentrations of BAs were significantly higher than 80 mg/L mentioned in the introduction?

R5: Thanks for your comments. The level of biogenic amines (BAs) in rice wine was greatly affected by the composition and metabolism of microbial flora in fermentation starter and brewing process, especially the growth and metabolism of BAs-producing bacteria. Previous studies have shown that Hongqu is a starter made by traditional technology in an open environment, and many factors may affect the microbial composition (including the amine-producing bacteria) of the final starter. So the flavor quality and BAs content of Hongqu rice wine brewed with Hongqu from different origin or batches are significantly different. This would explain the obvious differences in the content of BAs in Hongqu rice wine reported by different studies. In the future research, more fermentation starters (Hongqu) should be collected to study the microbial dynamics and flavor quality formation in Hongqu rice wine brewing.

Q6: The results of volatile flavor components were tested and analyzed for 45 days, while the results of metagenomic changes were tested only for 30 days. Did the authors analyze the correlation between metabolism and microbiome on Day 30?

R6: Thank you for the very good question raised by the reviewer. In this study, we systematically analyzed the change trend of physical and chemical parameters, biogenic amines (BAs) and volatile components in the brewing process of Hongqu rice wine (from day 1 to day 45), and found that these parameters remained basically constant after 30 days of brewing (parameters did not change obviously between day 30 and day 45). This may be mainly due to the lack of nutrients in the fermentation system and the restriction of ethanol produced by yeasts. In this study, we paid more attention to the formation process of volatile flavor components and BAs and their microbial regulatory pathways. Therefore, we selected six fermentation time points (day 1, 3, 5, 10, 20 and 30) during the brewing of Hongqu rice wine to analyze the composition and functional genes of the microbial flora according to the change trend of physicochemical parameters, BAs and volatile components. Of course, we will take your suggestions and improve them in subsequent meta-transcriptomic as well as meta-proteomic studies on Hongqu rice wine brewing.

Q7: I would suggest showing only the top 10 or 15 bacteria genres or species relative abundance in Figures 5 and 6. Only keep a few bacteria or fungi with high relative abundance in Figures 5 and 6 E and F.

R7: Thanks for your suggestions. In presenting the bacterial and fungal flora composition of the two Hongqu rice wines, HBAs and LBAs, we presented the top 30 bacterial and fungal genera and species in relative abundance. In addition, it has been reported that not only the dominant bacteria and fungi are the determinants of the flavor quality of fermented products, but also many non-dominant microorganisms play a crucial role in the maintenance of microecological functions and the formation of flavor quality of fermented products. We believe that our presentation of the microbial composition and the diversity of microbial species between HBAs and LBAs does not affect the final conclusions of this study. On the contrary, the presentation of some non-dominant microbial species will be more comprehensive.

Q8: The metagenomic sequencing method was used in this study, as the authors described in the introduction its features, I would recommend that the genus analysis results could be placed in the supplementary file.

R8: Thanks for your valuable advice. We have adjusted and modified this section according to your proposal.

Figure S1. Analysis of microbial genera in HBAs and LBAs brewing. The relative abundance of the predominant bacterial (A) and fungal (B) genera. Principal component analysis (PCA) of bacterial (C) and fungal (D) communities. Visualization of the differences in the relative abundance of bacterial (E) and fungal (F) genera between HBAs and LBAs. Microbial genera with significant differences between HBAs and LBAs were determined using a Welsh's t-test, and the Benjamini-Hochberg procedure was used to control the false-discovery rate due to multiple testing. Corrected P values are shown at right.

Q9: In this study, time series data were obtained and it was possible to analyze the relationship between changes in the group's microbiomes and BAs.

R9: Thanks for your valuable advice. The relationship between microbiome changes and BAs was analyzed in Section 3.5 in this paper, and key BAs were highlighted in the figure. However, I were sorry that the subtitle of section 3.5 may not be comprehensive enough, which brings inconvenience to readers.The correlation between the microbiome and BAs has been analyzed in Section 3.5 of this manuscript, and the BAs detected in Hongqu rice wine was highlighted in blue in Figure 6. We are sorry that the description in the manuscript submitted before may not be accurate enough, which may cause inconvenience to the reviewer.

Figure 6. Correlation analysis between the characteristic volatile components, biogenic amines (BAs) and the predominant microbial phylotypes at the species level during the brewing of Hongqu rice wine.

Q10: Statistical information is required in the figure legend, as in Figures 1, 2, and 4.

R10: Thanks for your valuable advice. We are very sorry that we did not describe this clearly in our previous submission, and in fact we did statistically analyze the experimental results. Statistical analysis of the data was performed using Statistical Package for the Social Sciences (SPSS) 23.0 (IBM, Armonk, New York). Student’s t-test was used for two-group comparisons (*p < 0.05, **p < 0.01). In Figure 4, we compared the difference in abundance of volatile flavor components in LBAs and HBAs through fold change chart. Log2 (LBAs/ HBAs) value greater than 1 or less than -1 combined with a p-value of less than 0.05 for one-way ANOVA indicates a significant difference in the abundance of this volatile substance.

Q11: Could authors upload the raw sequencing data to NCBI or other platforms? Because public sharing of sequencing data is an important part of microbiome research.

R11: Thanks for your valuable advice. We have uploaded the raw sequencing data and published it on the NCBI platform, and the accession number is PRJNA995035, which has been added in Section 2.7 of the revised manuscript.

2.7 Microbial DNA extraction and metagenomic analysis: “......Metagenomic sequencing data has been uploaded to the NCBI Sequence Read Archive (SRA) database (accession number: PRJNA995035).”

Round 2

Reviewer 3 Report

Thank you very much for the authors’ response and modifications to the paper! After carefully reviewing the authors’ response and the revised manuscript. I found that the authors have responded to my concern and I suggest this manuscript could be perfect.